# Planar Hall effect and anisotropic magnetoresistance in polar-polar interface of LaVO$_3$-KTaO$_3$ with strong spin-orbit coupling

Neha Wadehra [1], Ruchi Tomar[1], Rahul Mahavir Varma [2], R.K. Gopal[3], Yogesh Singh[3], Sushanta Dattagupta[4] & S. Chakraverty[1✉]

Among the perovskite oxide family, KTaO$_3$ (KTO) has recently attracted considerable interest as a possible system for the realization of the Rashba effect. In this work, we report a novel conducting interface by placing KTO with another insulator, LaVO$_3$ (LVO) and report planar Hall effect (PHE) and anisotropic magnetoresistance (AMR) measurements. This interface exhibits a signature of strong spin-orbit coupling. Our experimental observations of two fold AMR and PHE at low magnetic fields (**B**) is similar to those obtained for topological systems and can be intuitively understood using a phenomenological theory for a Rashba spin-split system. Our experimental data show a $B^2$ dependence of AMR and PHE at low magnetic fields that could also be explained based on our model. At high fields (~8 T), we see a two fold to four fold transition in the AMR that could not be explained using only Rashba spin-split energy spectra.

[1] Nanoscale Physics and Device Laboratory, Institute of Nano Science and Technology, Phase-10, Sector-64, Mohali, Punjab 160062, India. [2] Solid State and Structural Chemistry Unit, Indian Institute of Science, Bangaluru, Karnataka 560012, India. [3] Indian Institute of Science Education and Research Mohali, Knowledge City, Sector-81, SAS Nagar, Manauli 140306, India. [4] Bose Institute, P-1/12, CIT Rd, Scheme VIIM, Kankurgachi, Kolkata, West Bengal 700054, India. ✉email: suvankar.chakraverty@gmail.com

In recent times, the urge of attaining new functionalities in modern electronic devices has led to the manipulation of spin degree of freedom of an electron along with its charge[1,2]. This has given rise to an altogether new field of spin-electronics or "spintronics". It has been realized that momentum dependent splitting of spin-bands in an electronic system, the "Rashba effect", might play a key role in spintronic devices[3–5]. The Rashba effect is important not only because it might have tremendous potential for technical applications, but also because it is a hunting ground of emergent physical properties[6–14].

Semiconducting materials such as heterostructures of GaAs/GaAlAs and InAs/InGaAs have already been explored for the manifestation of the Rashba effect[15,16]. Another potentially rewarding class of materials for realization of this effect is "oxides"[17,18]. The benefit of using oxides for spin based electronic devices is that they manifest a wealth of functional properties like magnetoresistance, superconductivity, ferromagnetism, ferroelectricity, charge ordering etc. which can be coupled with the Rashba effect to achieve emergent phenomena if a suitable interface or superlattice is designed[19–22]. In addition to this,

simple cubic structure of perovskite oxides makes them easily usable for fabrication of heterostructures for device applications[17]. Also, in recent times oxide thin films and interfaces with strong spin-orbit coupling are predicted to exhibit topological phases owing to their non-trivial spin-structures and electronic states which may add further dimensions to the field of "oxide spintronics"[23].

Among perovskite oxides, $SrTiO_3$ (STO) has been widely explored for realization of 2DEG at its interface with other perovskite oxides such as $LaAlO_3$ (LAO), $LaVO_3$ (LVO), and $CaZrO_3$ (CZO) etc.[24–26]. However, the spin–orbit coupling strength of STO (which is a prerequisite for realization of the Rashba effect) is not very high. Another promising candidate from the perovskite oxide family having potential to host low-dimensional electron gas is $KTO$[27,28]. This insulating material has a dielectric constant and band gap similar to STO with an additional advantage of having strong spin–orbit coupling strength due to presence of 5d Tantalum atoms[29]. The energy level splitting in KTO due to SOC is ~400 meV which is an order of magnitude higher than that of STO (17 meV)[30].

With the aim of realizing 2DEG in a perovskite oxide with strong spin–orbit coupling, we grow heterostructure between LVO and KTO. The heterointerface is found to be conducting above the film thickness of three monolayers (ml). A carrier mobility of ~600 cm$^2$ V$^{-1}$ s$^{-1}$ is measured at the interface with varying thickness of the LVO film. Anisotropic magnetoresistance which is a relativistic magnetotransport phenomenon observed in magnetic and some topological systems is predicted in systems with the Rashba-Dresselhaus type spin-splitting[31–36]. We also report signature of topological chiral anomaly via observation of planar Hall effect and oscillations in longitudinal anisotropic magnetoresistance in our LVO-KTO system. A theoretical modeling using Rashba spin-split energy spectrum could predict our observation of 2-fold oscillations in AMR and PHE at low applied magnetic fields. The appearance of an additional periodicity in AMR above 8T magnetic field suggests a possible complex and rich physics arising from the interplay between chiral nature of the band structure, relativistic 2 dimensional itinerant electrons and strong spin–orbit coupling present in the system. We observe a $B^2$ dependence of AMR and PHE amplitudes which we are able to capture within our theoretical model.

## Results

**LVO-KTO interface.** Thin films of LVO were grown on (001) oriented Ta-terminated KTO single crystals using pulsed laser deposition (PLD) system (see details in Methods). The schematic of the heterostructure is shown in Fig. 1a. Different samples of varying thickness of LVO were grown. The thickness of the films was controlled using reflection high-energy electron diffraction (RHEED) technique. The RHEED oscillations of the specular spot, for 10 ml, 8 ml, 4 ml sample, as a function of number of unit cells are shown in Fig. 1b. Figure 1b also shows the RHEED pattern of the 10 ml sample before and after the film growth. Figure 1c shows the XRD plot of the 40 ml sample which confirms the crystalline growth of the LVO film. Inset of Fig. 1c shows the rocking curve of the KTO substrate depicting the high quality and crystallinity of the sample. We have also performed X-ray photoemission spectroscopy (XPS) of LVO thin film. The detail of the spectroscopy is discussed in supplementary information (Supplementary Note 3). Combining XRD, RHEED and XPS we confirm that high quality LVO is epitaxially formed on KTO substrate.

**Electrical properties.** Figure 2a shows the temperature dependent two dimensional resistivity ($\rho_{2D}$) for conduction parallel to the

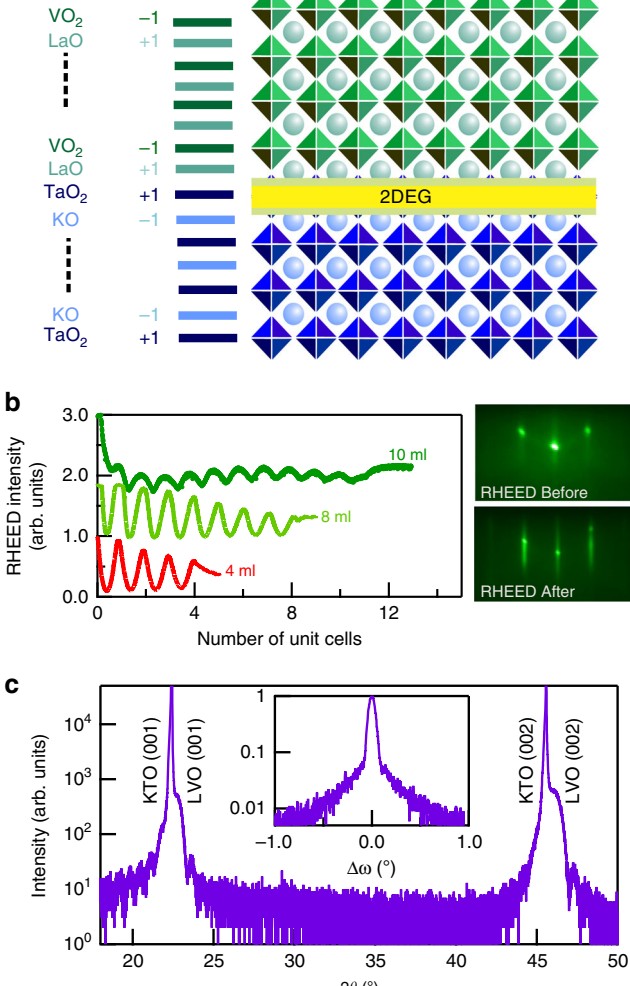

**Fig. 1 LVO-KTO interface structural characterization.** (Color online) **a** Schematic of the LVO-KTO heterostructure showing alternately charged layers in both LVO and KTO leading to formation of 2DEG at the interface (**b**) RHEED oscillations for 4, 8, and 10 ml LVO-KTO samples and RHEED pattern for 10 ml sample before and after growth of LVO film. **c** X-ray diffraction pattern of 40 ml sample showing crystalline film growth of LVO on KTO.

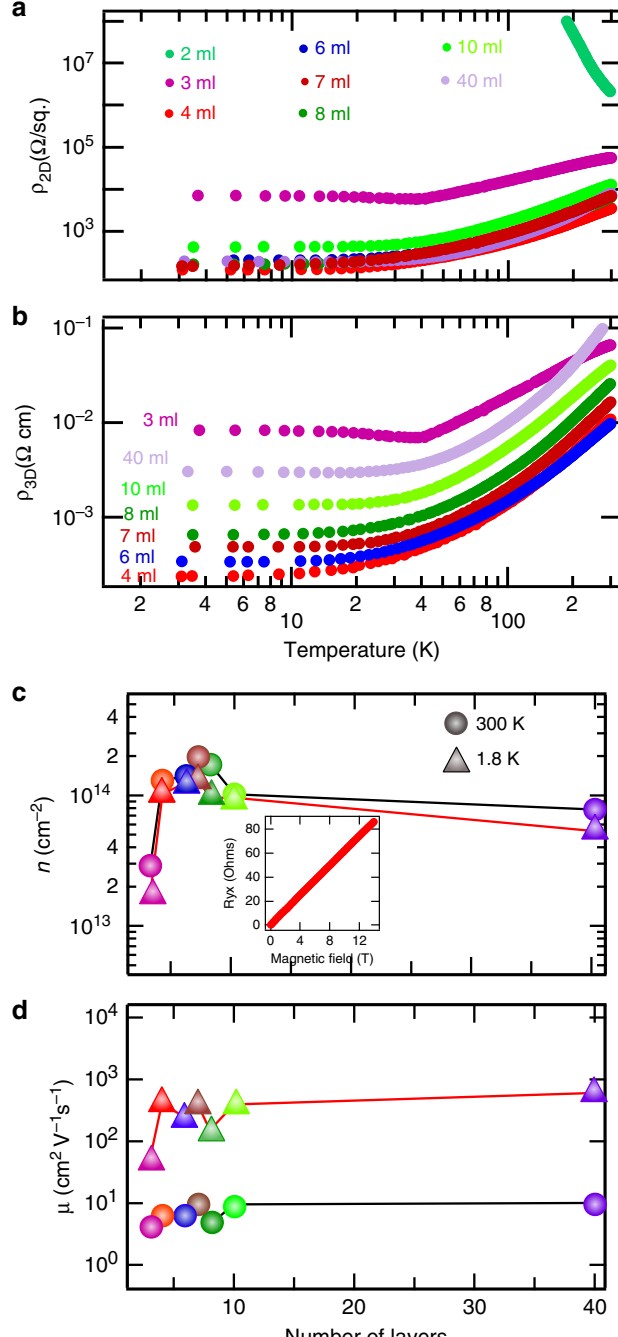

**Fig. 2 Electrical properties.** (Color online) **a** Temperature dependent 2D resistivity and **b** 3D resistivity for LVO-KTO samples with varying LVO thickness. **c** Charge carrier density and **d** mobility of all the samples measured at 300 K and 1.8 K. Inset of **b** shows the Hall data for 4 ml sample at 1.8 K.

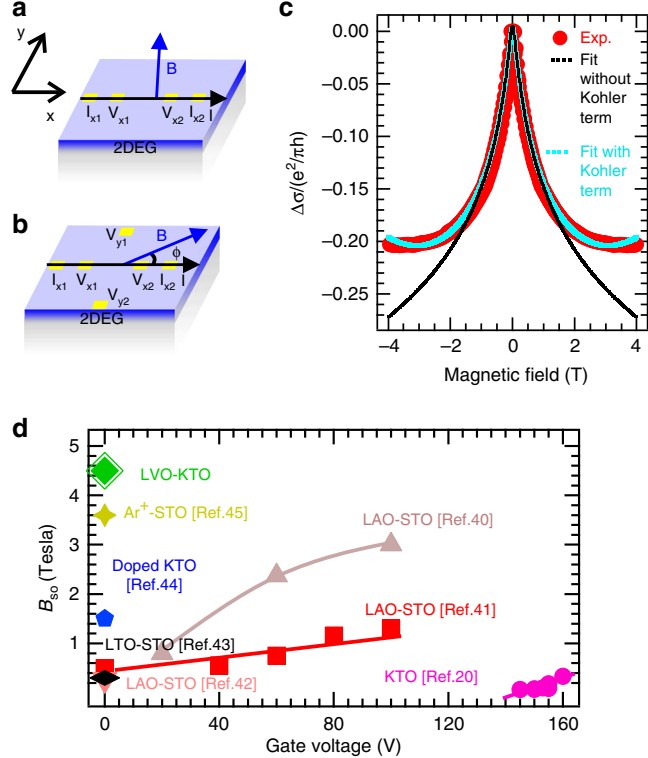

**Fig. 3 Spin–orbit coupling strength analysis.** (Color online) Schematic of the connection geometry for magnetoresistance ($R_{xx}$) measurements for magnetic field applied (**a**) out-of-plane and **b** in the plane. **c** Magnetoconductance plot of 4 ml sample as a function of magnetic field showing weak anti-localization due to high spin–orbit coupling along with the fitting done without the Kohler term (black line) and the fit including the Kohler term (cyan line). **d** Comparative plot of $B_{SO}$ vs. gate voltage for STO and KTO based systems.

conducting samples, calculated from the conventional Hall measurements done at 300 K and 1.8 K. It can be seen that above 3 ml of LVO, once the interface becomes conducting, the charge carrier density and mobility are independent of LVO thickness. This is in accordance with the electronic reconstruction mechanism for formation of 2DEG where after achieving the critical thickness to avoid polar catastrophe, increasing the thickness of the film does not add further carriers at the interface[25]. The charge carrier density of the samples can in principle be tuned using electrostatic gating and multicarrier physics can be explored but we plan to do it in a separate study. In the present case the linear variation of Hall resistance with magnetic field confirms the single type of charge carrier in our system. Figure 2b inset shows the Hall resistance of 4 ml LVO-KTO. The Hall data for other samples is presented in Supplementary Fig. 3. We obtained carrier mobility of ~600 cm² V⁻¹ s⁻¹ at 1.8 K in our samples as shown in Fig. 2d.

**Out-of-plane magnetotransport measurements.** The low temperature conventional magnetoresistance (MR) measurements where magnetic field is applied perpendicular to the interface of LVO and KTO (see Fig. 3a) on 4 ml sample reveals the presence of weak-antilocalization and hence strong spin–orbit coupling in the system[30,37,38]. Theory had been developed by Iordanskii, Lyanda-Geller, and Pikus (ILP theory) to describe the weak-antilocalization in magnetoconductance for the materials with strong spin–orbit coupling. The expression of the magnetoconductance developed by

interface and Fig. 2b shows the three dimensional resistivity ($\rho_{3D}$) normalized by the film thickness for all the samples. The 3-ml sample although conducting at room temperature exhibits an upturn near 30 K. All other samples with LVO more than 3 ml are conducting down to 1.8 K. In contrast to the wide range of values found for $\rho_{3D}$, the same data plotted as $\rho_{2D}$ shows that the data for all conducting samples essentially collapse to a narrow range of $\rho_{2D}$ values. This indicates that LVO film itself is indeed insulating and that only the interface forms the conducting channel. Figure 2c, d show the charge carrier density and mobility, for the

ILP theory is given by refs. [38,39].

$$\Delta\sigma = \frac{e^2}{2\pi^2\hbar}\left[ln\left(\frac{B_\phi}{B}\right) - \psi\left(\frac{1}{2} + \frac{B_\phi}{B}\right) + ln\left(\frac{B_{SO}}{B}\right)\right.$$
$$\left. - \psi\left(\frac{1}{2} + \frac{B_{SO}}{B}\right) - A_k\frac{\sigma_o}{G_o}B^2\right] \qquad (1)$$

where, $\mathbf{B}$ is the applied magnetic field, $B_\phi$ $(\hbar/4el_\phi^2)$ and $B_{SO}$ $(\hbar/4el_{SO}^2)$ are two characteristic magnetic fields related to phase coherence length ($l_\phi$) and spin-precession length ($l_{SO}$), $\psi$ is the digamma function and $G_o$ $(e^2/\pi h)$ is the quantum conductance. The ILP theory was derived for the magnetic field region $B < \hbar/2el_m^2$; where $l_m$ is the mean free path of the carriers[30,38]. For the present sample $\hbar/2el_m^2$ is estimated to be 0.3 T. However, we have been able to fit our data upto 1 T as shown in Fig. 3c fitting with black line. The last term with Kohler coefficient $A_k$ accounts for the orbital magnetoconductance having $B^2$ dependence. Figure 3c fit with cyan line shows the magnetoconductance data for 4 ml sample using full Eq. (1). A high value of $B_{SO} \sim$ 4.4 T corresponding to a spin-precession length of 6 nm was obtained from the fitting. Phase coherence length of 70 nm and magnetic field strength corresponding to inelastic scattering $B_\phi = 0.03$ T was obtained for our system. These values of phase coherence length and $B_\phi$ are in excellent agreement with the previous report[30].

In Fig. 3d, we have plotted the $B_{SO}$ of STO and KTO based systems as a function of applied gate voltage from the literature and compared it with our sample[30,40–45]. Figure 3d clearly suggests that our LVO-KTO interface has the highest $B_{SO}$ among all reported STO and KTO systems.

**PHE and AMR**. Figure 3b shows a schematic diagram of PHE and AMR measurement configuration, where magnetic field is applied in the sample plane and transverse resistance ($R_{yx}$) and longitudinal resistance ($R_{xx}$) are measured. Usually, PHE and AMR are observed in magnetic systems and are associated with the crystalline anisotropy of the system[31,32]. Also, recently some topological systems have been reported to witness in-plane AMR and PHE[33,34], the origin of which is anisotropic spin flip transition probabilities arising from broken time reversal symmetry. Theoretically, it has been predicted that the systems with the Rashba–Dressalhaus type of spin band splitting in presence of magnetic impurities may also exhibit in-plane AMR and PHE[35,36]. Although theoretically predicted, experimental realization of such phenomena in the 2DEG systems with high spin-orbit interaction is not well explored. Considering the large spin–orbit coupling obtained for our system, we expect interesting in-plane AMR and PHE, as well as their evolution as a function of the applied magnetic field.

For these measurements, magnetic field ($\mathbf{B}$) was applied in the sample plane and simultaneous measurements of longitudinal magnetoresistance ($R_{xx}$) and transverse resistance ($R_{yx}$) were made while varying the in-plane angle between $I$ and $\mathbf{B}$. For the first set of experiments, $R_{xx}$ and $R_{yx}$ were measured at 1.8 K by varying the magnitude of applied magnetic field. On scanning the angle between $\mathbf{B}$ and $I$, $R_{xx}$ and $R_{yx}$ were found to show oscillatory behavior. Upto 8 T, we obtained 2-fold periodic oscillations in $R_{xx}$, it slowly changed to 4-fold oscillations above 8 T. Figure 4a shows the $R_{xx}$ behavior at 3 T and 14 T. The behavior of normalized $R_{xx}$ on varying the applied magnetic field is shown in the contour plot presented in Fig. 4c, where $R_{xx}^{norm} = (R - R_{symm})/(R_o - R_{symm})$. $R_{symm} = R_{min} + (R_{max} - R_{min})/2$ and $R_{min}$ is minimum value of $R_{xx}$, $R_{max}$ is maximum value of $R_{xx}$, and $R_o$ is the value of $R_{xx}$ at 0°. The low field behavior of $R_{xx}$ is very similar to that observed in topological insulator systems such as $Bi_{2-x}Sb_xTe_3$ thin films[33].

We observed oscillations in the planar Hall resistance value as a function of in-plane angle between $\mathbf{B}$ and $I$, with minima at 45° and maxima at 135° repeated at 180° interval. Field dependent measurements were also performed at 1.8 K. Figure 4d shows the contour plot of field dependent $R_{yx}$ as a function of angle between $\mathbf{B}$ and $I$ at 1.8 K. It was seen that, on decreasing the magnetic field, the amplitude of oscillations decreases but the nature of oscillations remains same throughout. Figure 4b shows the planar Hall resistance for 14 T and 3 T field. It is clear that the 2-fold periodicity is maintained even for high fields. Figure 4e shows the Rashba-split spin bands. Figure 4f shows the transition probabilities between different bands as a function of in-plane angle between $\mathbf{B}$ and $I$. They are discussed in detail in the theoretical section below.

Two to four fold transitions in AMR have been reported in STO but these transitions are much complicated and irregular[46–49]. Such transitions in STO were explained in terms of Liftshitz transitions arising from the topological change in Fermi surface in presence of intrinsic magnetization of STO. The mechanism of such a transition for our system is not currently understood.

In addition to this, in topological materials, PHE originates from the Adler-Bell-Jackiw (ABJ) chiral anomaly and nontrivial Berry curvature and is regarded as an evidence of the Dirac/Weyl cones in the band structure[33,50–55]. In contrast, metals and semiconductors with trivial band structure, are not expected to show PHE. PHE and AMR in topological systems can be mathematically represented as Nandy et al.[52],

$$R^{PHE} = \Delta R^{chiral}sin(\Phi)cos(\Phi) \qquad (2)$$

$$R_{xx} = R_{xx}^{\phi=90°} + \Delta R^{chiral}cos^2(\Phi) \qquad (3)$$

where, $\Delta R^{chiral}$ is the chiral contribution to the PHE resistance ($R_{yx}$) and $R_{xx}^{\phi=90°}$ is the absolute value of longitudinal magnetoresistance ($R_{xx}$) for $\phi = 90°$. It was theoretically shown that the amplitude of PHE and AMR of such topological materials should follow $B^2$ dependence[52].

The angular dependence of PHE and AMR of our system is very similar to that observed for topological systems as shown in Fig. 5a, b. To illustrate this, we have plotted the amplitude of PHE (left axis) and AMR (right axis) as a function of applied magnetic field in Fig. 5c. We see that the amplitude of both PHE as well as AMR exactly follows a $B^2$ dependence upto 9 T magnetic field. For larger fields, both deviate from $B^2$ dependence. It is worth noting that above this field, AMR no more follows $cos^2(\phi)$ dependence (Fig. 5b). At the same time, although the 2-fold symmetry of PHE is retained above 9 T, the amplitude of PHE no more remains $B^2$ dependent.

**Discussion**

The observed 2-fold oscillations in the AMR and PHE as well as low field $B^2$ dependence of AMR and PHE amplitude on application of an in-plane magnetic field could be intuitively understood on the basis of electronic transitions which take place between the Rashba-split energy bands. In LVO-KTO system, due to broken inversion symmetry at the interface and subsequently developed electric field, the relativistic electrons in 5d orbitals of Ta experience a pseudo magnetic field in the conduction plane and hence may lead to Rashba spin-splitting. The occurrence of a significant spin-orbit interaction has already been reported in the literature, from ARPES measurements in a single crystal of KTO[44]. The presence of a Rashba spin-splitting, that relies on the additional presence of an electric field, was also seen in this material, for a Fermi wave vector ($\sim 0.2$ Å$^{-1}$ to 0.4 Å$^{-1}$) at a carrier density of $\sim 2 \times 10^{14}$ cm$^{-2}$. On the other hand, our system is not just KTO but its interface with LVO (a polar material). Hence, like KTO, the interface, for a (measured) carrier density of $1.02 \times 10^{14}$ cm$^{-2}$ at a (calculated) Fermi vector of 0.3 Å$^{-1}$, is not only endowed with a non-zero

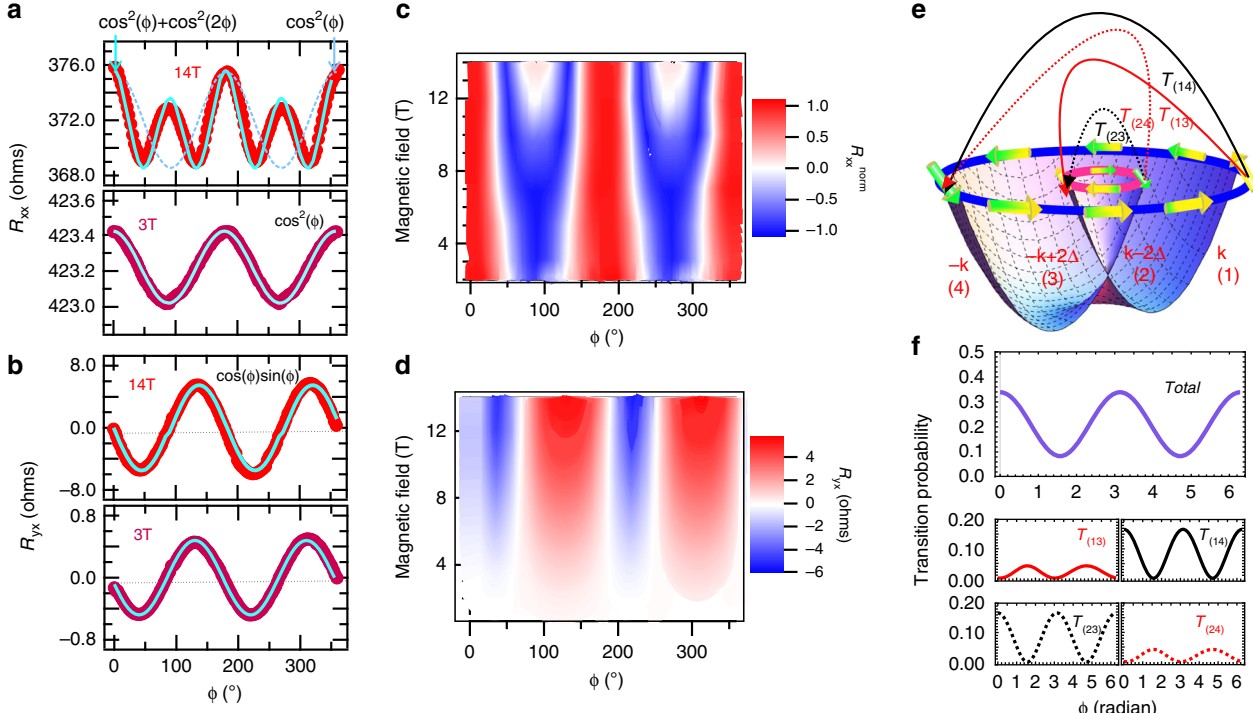

**Fig. 4 PHE and AMR measurements.** (Color online) **a**, **b** Angle dependent $R_{xx}$ and $R_{yx}$ measured at 1.8 K for 14 T and 3 T. Blue line is the fitted curve. **c**, **d** Applied magnetic field and angle dependent contour plots for normalized $R_{xx}$ and $R_{yx}$. **e** Rashba energy-split bands showing spin-texture at a particular energy and the allowed electronic transitions. **f** Total probability and individual probabilities for different allowed electronic transitions between the bands.

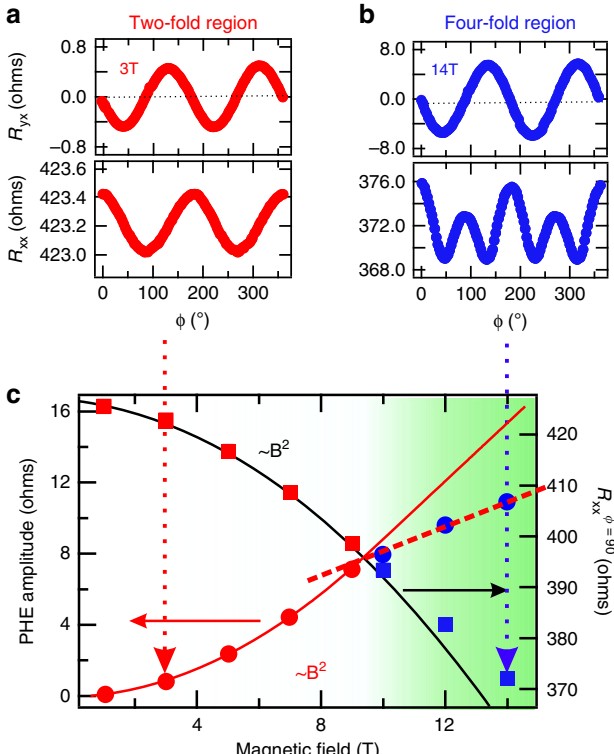

**Fig. 5 Signature of ABJ chiral anomaly.** (Color online) **a** $R_{yx}$ and $R_{xx}$ data for 3 T and **b** 14 T. **c** The red markers (white background area) show the low magnetic field region where AMR has two fold symmetry and blue markers (green background) show high-field region where AMR has 4-fold symmetry. Circles represent the PHE amplitude as a function of magnetic field and squares represent $R_{xx}^{\phi=90°}$ i.e. absolute value of magnetoresistance $R_{xx}$ for $\phi = 90°$ as a function of magnetic field.

spin–orbit coupling, but is also expected to exhibit a prominent Rashba effect in view of a substantial, polar–polar interface-generated electric field. Our analysis presented below, is based in this premise.

In our system, the degenerate energy parabola of electrons is Rashba split into two parabolas as shown in Fig. 4e. Application of an external magnetic field in the conduction plane further adds a Zeeman splitting term. The external parabola is called the majority band and the internal parabola is called the minority band. Depending on the propagation vector **k**, spin of the electron, Rashba strength parameter $\alpha$, and the direction and magnitude of the external applied magnetic field, the electrons can make transitions between majority-to-majority (or minority-to-minority) i.e. intra-band transitions and majority-to-minority (or minority-to-majority) i.e. inter-band transitions. Each allowed transition results in back-scattering of the conduction electrons and hence, contributes to increase in resistance. The energy eigen values for the spin bands can be calculated by solving the Hamiltonian, which in the absence of magnetic field, can be written as[35]:

$$H = \epsilon(k) - \alpha(\sigma_x k_y - \sigma_y k_x) \quad (4)$$

where, $\epsilon(k)$ is free electron energy, $\sigma_{(x, y)}$ are the Pauli spin matrices, and $k_x$ and $k_y$ are the wave vectors in **x** and **y** direction. The electronic transition probability between the bands can be calculated using the eigen vectors for each band and finding the transition matrices. The eigen vectors used for the majority and minority bands are:

$$\frac{1}{\sqrt{2}}\begin{pmatrix} 1 \\ ie^{i\theta k} \end{pmatrix} \text{ and } \frac{1}{\sqrt{2}}\begin{pmatrix} 1 \\ -ie^{i\theta k} \end{pmatrix}$$

respectively, where, $\theta$ is the angle between **k**-vector and **x**-axis. Figure 4e shows the Rashba energy-split bands (numbered as 1, 2, 3, and 4) with spin texture for a fixed energy value.

Now, imagine that a **B**-field is applied in a direction which is also coincident with the above **x**-axis (which however is distinct from

the **x**-axis in the laboratory frame i.e., the frame of the planar sample, along which the current is applied). The present **x**-**y**-frame then defines the principal coordinate system, in the sense of Taskin et al.[33]. The corresponding Zeeman term contains the x-component of the Pauli spin which does not affect the term proportional to $k_y$, in Eq. (4). However, the Zeeman coupling, being off-diagonal in the $\sigma_y$-representation, can cause a spin-flip thereby triggering a reversal in $k_x$. Interestingly, these momentum-reversal transitions are akin to Drude scattering that is ever present, albeit small, as a 'residual' resistivity[56]. Now, when the **B**-field is not too large (<8 T, in our experiments), the additional transition probabilities due to the Zeeman interaction can be calculated from the 'Golden Rule' of perturbation theory, thus yielding a quadratic dependence on **B**, which have to be supplanted to the residual Drude resistivity parallel to **x**-axis. The perpendicular component however retains only the residual part, for reasons mentioned earlier (for details see Supplementary Note 1).

The allowed transitions between different bands ($T_{13}$, $T_{14}$, $T_{23}$, $T_{24}$) having finite probability are shown with arrows in Fig. 4e. These have been computed in the supplementary section (Supplementary Note 2) and are presented in Fig. 4f. With the parallel and perpendicular ("diagonal") components in hand we can transform back to the laboratory frame a la Taskin et al.[33]. Since the residual Drude resistivity cancels out from the difference between the parallel and perpendicular components, the dominant contribution to the resistivity arises from the Rashba effect. Further, following Taskin et al.[33], $R_{xx} \sim (\cos^2\phi)$, while $R_{yx} \sim (\cos\phi \sin\phi)$, both being proportional to $B^2$, in conformity with our data shown Fig. 5. Here, $\phi$ is the angle between the applied magnetic field and the direction along which the current is measured. As we mentioned earlier, beyond 8 T, there is a substantial departure in the ($\cos^2\phi$)-behavior of $R_{xx}$, as well as its $B^2$ dependence, a theoretical understanding of which requires going beyond the Golden Rule of perturbation theory and perhaps also the simple Rashba effect, implemented in this paper.

In the present system, we speculate that it might be due to the relativistic character with strong spin–orbit coupling of the carriers in the system. Our observations suggest a detailed theoretical model of such systems is essential and it would have to contain ingredients of low dimensionality, relativistic electrons, localized magnetic moments, and strong spin–orbit coupling. Further evolution of the Fermi surface with electrostatic gating and in-plane magnetic field can be explored which we plan to do in a follow-up study to build up a complete understanding of the mechanism[57].

In conclusion, we have realized a high mobility two dimensional electron gas at a new interface of two polar-polar perovskite oxides. We have observed a high spin–orbit coupling in the system. The magneto-transport measurements show signature of in-plane anisotropic transverse and longitudinal magnetoresistance as a consequence of strong spin–orbit coupling and Rashba spin splitting. The observed nature of the AMR and PHE at low magnetic field show very similar behavior as observed in topological materials having Weyl-fermions due to ABJ chiral anomaly. The observed features of PHE and AMR at low magnetic field could be understood from our phenomenological theory with Rashba-spin splitting. The high field four fold AMR warrants an elaborate theoretical analysis. Such a model system may open up an avenue for in depth understanding of the physical properties of low dimensional relativistic electrons in oxide materials with strong spin–orbit coupling. Such detailed understanding might play an important role in the design of new materials for spintronic applications.

## Methods

**Substrate preparation**. Thin films of LVO were grown on (001) oriented Ta-terminated KTO single crystals. For Ta-termination, method of high temperature annealing followed by DI water etching was employed[27]. The KTO (001) single crystals were annealed at an optimized temperature of 650 °C for 2 h in air under ambient conditions. Two substrates were annealed at one time with one kept upside down at the top of other with a gap of 60 μm using two sapphires on the sides. This was done to avoid K vacancies. To anneal, the substrates the ramp rate was kept 300 °C/hour while heating and it was kept 250 °C/hour while cooling down to room temperature. The annealing accumulated KO particles on the surface of the substrate which were removed by etching with deionised (DI) water heated at 60 °C giving us $TaO_2$ terminated step and terrace like structure.

**Thin film growth**. The thin films of LVO were grown using pulsed laser deposition (PLD) system. Polycrystalline $LaVO_4$ was used as the target material and was ablated using KrF excimer laser at a frequency of 2 Hz[25,58]. The laser fluence for target ablation was optimized to be 4 Jcm$^{-2}$. During growth, the substrate was kept at an optimized temperature of 600 °C using IR laser heating system and oxygen partial pressure of the deposition chamber was maintained to be $1 \times 10^{-6}$ Torr. The thickness of the films was controlled using reflection high-energy electron diffraction (RHEED) technique.

**Transport measurements**. The electrical as well as magneto-transport properties of the grown heterostructures were measured using physical property measurement system (PPMS) (Quantum Design, Dynacool setup, 14 T). For these measurements, contacts were made by ultrasonically wire-bonding the interface in either four probe or Hall geometry. The angle between the sample plane or the current and magnetic field was varied using horizontal rotator of PPMS.

## Data availability
The data that support the findings of this study are available from the corresponding author upon reasonable request.

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

## Acknowledgements

We acknowledge Prof. D.D. Sarma of Indian Institute of Science, Bangaluru, India, for the XPS meaurements and related discussion. N.W. and S.C. acknowledge Prof. Pushan Ayyub of Tata Institute of Fundamental Research, Mumbai, India for helpful discussion. The financial support from Department of Science and Technology (DST), India - Nano Mission project number (SR/NM/NS-1007/2015) is acknowledged. S.D. is grateful to the Indian National Science Academy for support through their Senior Scientist scheme and to INST for hospitality.

## Author contributions

N.W. synthesized and characterized the heterostructrues and did theoretical simulations. R.T. helped in growth condition optimization. R.M.V. did XPS measurements. N.W. and R.K.G. performed the anti-localization fitting. Y.S. helped in PHE and AMR analysis. S.D. built up the theoretical model. S.C. designed and supervised all the experiments. N.W. and S.C. wrote the paper. All authors discussed the data and commented on the paper.

## Competing interests

The authors declare no competing interests.
