## [Peer Review File · Nature Communications]

Reviewers' comments:

Reviewer #1 (Remarks to the Author):

Wadehra et al. reported the planar Hall effect and anisotropic MR in LaVO₃/KTaO₃ two-dimensional electron gas. They attributed their observations to the Rashba-spin splitting and built a theoretical model to describe the 2-fold to 4-fold transition in MR. To my understand, the topic itself is a bit out-of-fashion, the materials choices are not new, the experimental results are lack of solid theoretical explanation. I could not support it to be published in Nature Communications. The reasons are listed below in detail,

- 1) Magnetic 2DEG should have lots of interest in developing spintronic high-mobility electronic devices. However, the low-temperature, small response, and relative low mobility (compared to semiconducting 2DEG) limit their wide applications. So far, only few groups are still digging down in such a topic, like Pyrids in Danmark, and so on. To say that, I did not see any progresses in the present work to push this topic forward in both experiments and theoretical understanding.
- 2) Recently, Sun and Han groups had reported the 2 DEG at the EuO-KTO interfaces (PRL 121, 116803 (2018) and Nano Lett. 19, 1605-1612 (2019)). Using 5d element Ta in KTO to enhance the spin-orbital coupling is not a new approach. The LaVO₃ is known to be difficult to grow and needs a reduced environment. Therefore, how to get an non-oxygen deficient interface should be addressed. More characterizations to proof the valence states, interface qualities, magnetic distributions, and etc should be provided at minimum requirements.
- 3) The authors claimed there might be uncompensated V spins at the interfaces. The arguments are quite doubted and lack of necessary evidences. I pointed this out because this point is the only weight-point in this paper, unfortunately, still not fully convincing. This makes the quality of this paper degraded. Fig. 4(e) and 4(f) have lots of speculations. The theoretical model is too simple and not supported by their results.
- 4) Actually, except for the Fig. 4, the other three figures are simply the basic characterizations. I did not learn anything new from these results. Perhaps, these should be put into the supporting information. And the authors should focus only on the Figure 4 and expand the explanations, provide more solid evidence, fit their results with their model, etc.

Reviewer #2 (Remarks to the Author):

The manuscript "Planar Hall effect and Anisotropic Magnetoresistance in a polar-polar interface of LaVO₃-KTaO₃ with strong spin-orbit coupling", by Neha Wadehra, et al., seems to contain results that are sound and interesting enough to warrant publication.

Before a final decision is made, I ask the authors to consider the following comments and suggestions, to improve the quality of their presentation.

1. The list of evidences for emergent physical properties of Rashba systems, should be updated somewhat, after 2013. I found some Refs.

Sci. Rep. DOI: 10.1038/srep12751;

EPL, 112 (2015) 17004 www.epjjournal.org doi: 10.1209/0295-5075/112/17004;

Nat. Mater. 15, 1261–1266 (2016);

Nat. Mater. 15, 1224–1225 (2016);

PRL 119, 256801 (2017);

but the authors may find other references too.

2. The concept of 2D and 3D resistivity is confusing, I rather suggest to discuss it in term of contributions parallel and perpendicular to the interface.

3. As I understand, the authors did not attempt a study as a function of the carrier density, so the reader is left wondering whether multiband/multicarrier physics may play a role at these interfaces, like it does in LAO/STO (see, e.g., <https://doi.org/10.1038/s41563-019-0354-z>). I was wondering whether the authors can at least discuss this issue, or plan future studies. This issue is particularly relevant, because the authors measure the Hall effect up to high magnetic fields.

4. Can the authors explain why there is no need for a Kohler contribution in Eq. (1)? Such a contribution is expected in high-mobility metallic systems (see, e.g., DOI: 10.1038/srep12751).

5. I found two occurrences of cm^2 rather than cm^{-2} for the units of the carrier density.

6. The effect of an external magnetic field on the Rashba band structure has been discussed in PRB 89, 195448 (2014).

Once the authors have complied with the above issues, I suggest that their manuscript may be accepted for publication.

Reviewer #3 (Remarks to the Author):

The authors report transport measurements in a new 2DEG stabilized at the KTO/LVO oxide interface, arising from a polar catastrophe analogous to the LAO/STO interface. The material quality looks very good given the RHEED oscillations indicating layer-by-layer growth, and the narrow rocking curve from XRD.

One of the major findings is the 4.4 T spin-orbit field determined from weak antilocalization fits. This shows that the spin-orbit coupling in KTO/LVO is the largest amongst oxide 2DEGs as seen from Fig. 3c. This is potentially of great interest to the spintronics community. It would have been interesting to know how this changes with gating.

The authors also present interesting Planar Hall Effect (PHE) R_{xy} and Anisotropic Magnetoresistance (AMR) R_{xx} data as a function of the angle θ between the current flow direction and the in-plane field. At low fields (~ 3 T) they see the expected two-fold AMR going like $\cos^2(\theta)$, but at high fields (~ 14 T) it changes to a four-fold symmetry $\cos^4(\theta)$. Such a term is allowed by the crystal symmetry, but it is not described by the theoretical modeling presented here. The authors do not comment on the angular dependence of the PHE, which retains two-fold symmetry at all fields. Here are some ways in which the authors can improve their manuscript.

1) From the simple Rashba SOC model, the authors obtain the conventional $\cos^2(\theta)$ dependence

in R_{xx} in the low magnetic field regime. However, the authors neglect to explain if this model correctly predicts the angular dependence for R_{xy} in this regime. The authors should discuss this.

2) Looking at the calculation for R_{xx}^{norm} , I feel that there may be a minor mistake. From Fig. 4a, it seems that $R_{xx}^{\text{symm}} \sim 372$ Ohms. The value of R_0 (R_{xx} at 0 degrees) is ~ 376 Ohms. So, R_{xx}^{norm} at 0 degrees would be $(376-372)/376 = 0.01$. However, in Fig. 4c R_{xx}^{norm} changes from -1 to 1 as labeled in the color scale? Perhaps the corrected equation is

$$R_{xx}^{\text{norm}} = (R - R_{xx}^{\text{symm}}) / (R_0 - R_{xx}^{\text{symm}})$$

3) Since R_{xx}^{norm} is dimensionless, it should not have units of Ohms in the color bar Fig. 4c.

I feel that the modeling should be strengthened, but the experimental data at the core of the paper is compelling even if the field variation of the AMR symmetries is not yet well understood. On balance, I am happy to recommend publication in Nature Communications once the above questions are addressed.

Referee #1

Wadehra et al. reported the planar Hall effect and anisotropic MR in $\text{LaVO}_3/\text{KTaO}_3$ two-dimensional electron gas. They attributed their observations to the Rashba-spin splitting and built a theoretical model to describe the 2-fold to 4-fold transition in MR. To my understand, the topic itself is a bit out-of-fashion, the materials choices are not new, the experimental results are lack of solid theoretical explanation. I could not support it to be published in Nature Communications. The reasons are listed below in detail,

We disagree with the Referee's comment that the topic is "out-of- fashion" and the material isn't new. We report discovery of several novel magnetotransport behavior in the $\text{LaVO}_3\text{-KTaO}_3$ (LVO-KTO) interface and propose a model including Rashba coupling to explain many of the observed features. Both the other Referees have acknowledged the high quality of the interface, the large spin-orbit (SO) field for this interface which is the highest for any oxide 2DEG, the novelty of the results and their suitability for the journal after some revisions. In response to the Referee's comment below we reiterate a list of novel finding in this work.

1) Magnetic 2DEG should have lots of interest in developing spintronic high-mobility electronic devices. However, the low-temperature, small response, and relative low mobility (compared to semiconducting 2DEG) limit their wide applications. So far, only few groups are still digging down in such a topic, like Pyrids in Danmark, and so on. To say that, I did not see any progresses in the present work to push this topic forward in both experiments and theoretical understanding.

We thank the Referee for her/his comment. The Referee has correctly pointed out that 2DEG has lots of interest in developing spintronic high-mobility electronic devices but at the same time we would also like to point out that the main objective of this work is not to push the current limits of applicability of 2DEGs. We report several novel physical behavior in the LVO-KTO 2DEG which have not been reported before and some of which are not even expected in such systems. For the benefit of the Referee we list the novel findings of our study below: i) Highest SO coupling for any oxide 2DEG, ii) anomalous magnetotransport behavior like AMR and PHE which are normally expected in strongly magnetic, or in topological materials, iii) In the revised manuscript we include new observation of topological character in the electronic band structure similar to the systems having Dirac/Weyl fermion, the mechanism of whose appearance has to be novel and not known before.

The Referee's assertion ("still digging down in such a topic") is uncalled for. We disagree that only a few groups are currently working on these topics. To refute this, we list a few top groups, which are working on such topics. Many more researchers all over the world are active in this field. The list in alphabetical order, clearly suggests that researchers from world's leading institutions are working on this topic:

1. Prof. Akira Ohtomo
Department of Applied Chemistry,
Graduate School of Science and Engineering,
Tokyo Institute of Technology, Japan.
aohtomo@apc.titech.ac.jp
2. Prof. Chang-beom Eom
College of Engineering,
University of Wisconsin-Madison, USA.
eom@engr.wisc.edu
3. Prof. Charles Ahn
School of Engineering and Applied Science,
Yale University, USA.
charles.ahn@yale.edu
4. Dr. Chris Bell
School of Physics,
University of Bristol, USA.
christopher.bell@bristol.ac.uk
5. Dr. D.G Schlom
Department of Materials Science and Engineering,
Cornell University, USA.
Schlom@Cornell.edu
6. Prof. Dave H. A. Blank
Institute for Nanotechnology,
University of Twente, Netherlands.
d.h.a.blank@utwente.nl
7. Prof. Harold Y. Hwang
SLAC National Accelerator Laboratory,
Stanford University, USA.
hyhwang@stanford.edu
8. Prof. Ing Guus Rijnders
Inorganic Materials Science group,
University of Twente, Netherlands.
a.j.h.m.rijnders@utwente.nl
9. Prof. J. Fontcuberta
Magnetic Materials and Functional Oxides Group,
ICMAB-CSIC, Barcelona, Spain.
fontcuberta@icmab.es

10. Prof. Jean-Marc Triscone
Department of Quantum Matter Physics,
University of Geneva, Switzerland.
Jean-Marc.Triscone@unige.ch
11. Prof. Jeremy Levy
Department of Physics and Astronomy,
University of Pittsburg, USA.
jlevy@pitt.edu
12. Prof. Jirong Sun
Department of Condensed Matter Physics,
Institute of Physics of the Chinese Academy of Sciences, China.
Jrsun@203.iphy.ac.cn
13. Prof. Jochen Mannhart
Department of Solid State Quantum Electronics,
Max Planck Institute of Solid State Research in Stuttgart, Germany.
j.mannhart@fkf.mpg.de
14. Dr. Kazunori Ueno
Department of Basic Sciences, Graduate School of Arts and Sciences,
The University of Tokyo, Japan.
ueno@phys.c.u-tokyo.ac.jp
15. Dr. Manuel Bibes
CNRS/Thales Lab,
Thales Research and Technology, France.
manuel.bibes@thalesgroup.com
16. Prof. Masashi Kawasaki
Strong Correlation Interface Research Group,
Center for Emergent Matter Science, RIKEN, Japan.
m.kawasaki@riken.jp
17. Prof. Mikk Lippmaa
Institute for Solid State Physics,
University of Tokyo, Japan.
lippmaa@issp.u-tokyo.ac.jp
18. Dr. M. Nakano
Department of Applied Physics, School of Engineering
The University of Tokyo, Japan.
nakano@ap.t.u-tokyo.ac.jp

19. Prof. Paolo Radaelli
Department of Physics,
University of Oxford, UK.
charles.ahn@yale.edu
20. Prof. Pu Yu
Department of Physics,
Tsinghua University, China.
yupu@mail.tsinghua.edu.cn
21. Prof. Stuart S.P. Parkin
Max Planck Institute of Microstructure Physics, Germany.
stuart.parkin@mpi-halle.mpg.de
22. Prof. Susanne Stemmer
Materials Research Lab,
University of California, Santa Barbara, USA.
stemmer@mrl.ucsb.edu
23. Prof. T. Fukumura
Inorganic Solid State Chemistry Laboratory,
Tohoku University, Japan.
tomoteru.fukumura.e4@tohoku.ac.jp
24. Prof. T Venky Venkatesan
Department of Physics,
National University of Singapore, Singapore.
venky@nus.edu.sg
25. Dr. Yasuyuki Hikita
Stanford Institute for Materials and Energy Sciences,
Stanford University, USA.
hikita@slac.stanford.edu
26. Prof. Yuri Suzuki
Department of Applied Physics,
Stanford University, USA.
ysuzuki1@stanford.edu

2) Recently, Sun and Han groups had reported the 2 DEG at the EuO-KTO interfaces (PRL 121, 116803 (2018) and Nano Lett. 19, 1605-1612 (2019)). Using 5d element Ta in KTO to enhance the spin-orbital coupling is not a new approach. The LaVO₃ is known to be difficult to grow and needs a reduced environment. Therefore, how to get a non-oxygen deficient interface should be

addressed. More characterizations to proof the valence states, interface qualities, magnetic distributions, and etc should be provided at minimum requirements.

We agree with the Referee that 5d materials have been used in the past to enhance SO coupling. However, our material shows several novel features not seen for other such heterostructures. For example i) our estimated SO field is the highest reported for any oxide 2DEG, ii) our LVO-KTO interface shows novel magnetotransport behavior arising from strong SO coupling and Rashba effects, iii) Our interface shows behavior consistent with topological character of charge carriers.

To discuss the quality of LaVO₃ (LVO) film and interface, we would like to mention that we have performed environment optimization process to realize the best growth parameters of LVO. The extremely high quality of the film and the interface is confirmed from the following points (it is nice of the Referee 3 to have specifically mentioned about the high quality of our film and interface):

- a) The observed oscillations in RHEED intensity suggest a layer-by-layer growth.
- b) Sharp rocking curve obtained from the X-ray diffraction suggests high crystallinity of LVO film.
- c) Laue fringes have been observed very clearly in the X-ray diffraction pattern suggest an atomically abrupt and well defined interface.

Oxygen deficient LVO would exhibit enhanced conductivity. Our measurements of 3D conductivity of LVO-KTO films with varying LVO thickness do not scale with LVO thickness. This clearly demonstrates that only the interface is conducting and not the LVO layer itself. This strongly indicates the absence of oxygen vacancies in LVO films. Also, here we would like to point out that our substrate is KTaO₃ (KTO) and unlike SrTiO₃ it is extremely difficult to reduce KTO. Achieving oxygen vacant KTO by annealing at the growth conditions which we use to grow LVO is not possible, we have checked this possibility.

3) The authors claimed there might be uncompensated V spins at the interfaces. The arguments are quite doubted and lack of necessary evidences. I pointed this out because this point is the only weight-point in this paper, unfortunately, still not fully convincing. This makes the quality of this paper degraded. Fig. 4(e) and 4(f) have lots of speculations. The theoretical model is too simple and not supported by their results.

We thank the Referee for this comment. We point out that the presence of uncompensated V spins was a speculation, not backed by evidence. The presence of V-spins is not really required to explain our observed PHE and AMR, and we have now removed such statements from the revised manuscript. Rather we would like to point out that in the revised manuscript, we have added further analysis of our data and the new figure: Fig. 5 suggests that our observed PHE and AMR may be related to Adler-Bell-Jackiw (ABJ) chiral anomaly and topology of the band structure. We have accordingly modified the text as well.

Fig.4 (e) and (f) give a minimal but realistic model to help explain at least some of our results. Infact the model explains the low field magnetotransport data excellently. We have already

mentioned above that we have now included further analysis of our data which suggests chiral (topological) contribution to the magnetoresistance. This adds to the novel results discovered in this work.

We reproduce the text related to our extended data analysis along with new references that has been added on page 4, column 2, last paragraph:

“In addition to this, in topological materials, PHE originates from the Adler-Bell-Jackiw (ABJ) chiral anomaly and nontrivial Berry curvature and is regarded as an evidence of the Dirac/Weyl cones in the band structure.[33, 50-55] In contrast, metals and semiconductors with trivial band structure, are not expected to show PHE. PHE and AMR in topological systems can be mathematically represented as,[52]:

$$R^{\text{PHE}} = \Delta R^{\text{chiral}} \sin(\varphi) \cos(\varphi)$$

$$R_{xx} = R_{xx}^{\varphi=90} + \Delta R^{\text{chiral}} \cos^2(\varphi)$$

where, ΔR^{chiral} is the chiral contribution to the PHE resistance (R_{yx}) and $R_{xx}^{\varphi=90}$ is the absolute value of longitudinal magnetoresistance (R_{xx}) for $\varphi=90^\circ$. It was theoretically shown that the amplitude of PHE and AMR of such topological materials should follow B^2 dependence.[52]

The angular dependence of PHE and AMR of our system is very similar to that observed for topological systems as shown in Fig. 5(a). To illustrate this, we have plotted the amplitude of PHE (left axis) and AMR (right axis) as a function of applied magnetic field in Fig. 5 (b). We see that the amplitude of both PHE as well as AMR exactly follows a B^2 dependence upto 9 T magnetic field. For larger fields, both deviate from B^2 dependence. This is worth noting that above this field, AMR no more follows $\cos^2(\phi)$ dependence (Fig. 5(a)). At the same time, although the two fold symmetry of PHE is retained above 9T, the amplitude of PHE no more remains B^2 dependent.”

References:

[50] R. Singha, S. Roy, A. Pariari, B. Satpati, and P. Mandal, Phys. Rev. B 98, 081103(R) (2018).

[51] A.A. Burkov, Phys. Rev. B 96, 041110(R) (2017).

[52] S. Nandy, G. Sharma, A. Taraphder, and S. Tewari, Phys.Rev. Lett. 119, 176804 (2017).

[53] N. Kumar, S.N. Guin, C. Felser, and C. Shekhar, Phys.Rev. B 98, 041103(R) (2018).

[54] S. Nandy, A. Taraphder, and S. Tewari, Sci. Reports 8,14983 (2018).

[55] D. Ma, H. Jang, H. Liu, and X.C. Xie, Phys. Rev. B 99,115121 (2019).

Then on page 5, column 2, paragraph 3, line 1, we have inserted:

“In our system, the degenerate energy parabola of electrons is Rashba split into two parabolas as shown in Fig. 4(e). Application of an external magnetic field in the conduction plane further adds

a Zeeman splitting term. The external parabola is called the majority band and the internal parabola is called the minority band. Depending on the propagation vector k , spin of the electron, Rashba strength parameter α and the direction and magnitude of the external applied magnetic field, the electrons can make transitions between majority-to-majority (or minority-to-minority) i.e. intra-band transitions and majority-to-minority (or minority-to-majority) i.e. inter-band transitions. Each allowed transition results in back-scattering of the conduction electrons and hence, contributes to increase in resistance. The energy eigen values for the spin bands can be calculated by solving the Hamiltonian, which in the absence of the applied magnetic field, can be written [35]:

$$H = \epsilon(k) - \alpha(\sigma_x k_y - \sigma_y k_x) \quad (4)$$

Where, $\epsilon(k)$ is free electron energy, $\sigma(x; y)$ are the Pauli spin matrices, and k_x and k_y are the wave vectors in x and y direction. The electronic transition probability between the bands can be calculated using the eigen vectors for each band and finding the transition matrices. The eigen vectors used for the majority and minority bands are:

$$(1 \quad i e^{i\theta k})/\sqrt{2} \text{ and } (1 \quad -i e^{i\theta k})/\sqrt{2} ,$$

respectively, where, θ is the angle between the k -vector and the x -axis. Figure 4(e) shows the Rashba energy-split bands (numbered as 1, 2, 3 and 4) with spin texture for a fixed energy value.

Now, imagine that a B-field is applied in a direction which is also coincident with the above x -axis (which is however distinct from the x -axis in the laboratory frame, i. e., the frame of the planar sample, along which the current is applied). The present xy -frame then defines the principal coordinate system, in the sense of Taskin et al. [33] The corresponding Zeeman term contains the x -component of the Pauli spin which does not affect the term proportional to k_y , in Eq. (4). However, the Zeeman coupling, being off-diagonal in the σ_y -representation, can cause a spin-flip thereby triggering a reversal in k_x . Interestingly, these momentum-reversal transitions are akin to Drude scattering that is ever present, albeit small, as a ‘residual’ resistivity. [56] Now, when the B-field is not too large (< 8 T, in our experiments), the additional transition probabilities due to the Zeeman interaction can be calculated from the ‘Golden Rule’ of perturbation theory, thus yielding a quadratic dependence on B, which have to be supplanted to the residual Drude resistivity parallel to the x -axis. The perpendicular component however retains only the residual part, for reasons mentioned earlier.

The allowed transitions between different bands ($T_{13}, T_{14}, T_{23}, T_{24}$) having finite probability are shown with arrows in Fig. 4(e). These have been computed in the supplementary section and are presented in Fig. 4(f). With the parallel and perpendicular (‘diagonal’) components in hand we can transform back to the laboratory frame *a la* Taskin et al. [33] Since the residual Drude resistivity cancels out from the difference between the parallel and perpendicular components, the dominant contribution to the resistivity arises from the Rashba effect. Further, following Taskin et al. [33], $R_{xx} \sim (\cos^2 \varphi)$, while $R_{yx} \sim (\cos \varphi \cdot \sin \varphi)$, both being proportional to $(B)^2$, in conformity with our data shown in Fig. 5 below. Here, φ is the angle between the applied magnetic field and the direction along which the current is measured. As we mentioned earlier, beyond 8 T, there is a substantial departure in the $(\cos^2 \varphi)$ -behavior of R_{xx} , as well as its $(B)^2$ dependence, a theoretical

understanding of which requires going beyond the Golden Rule of perturbation theory and perhaps also beyond the simple Rashba effect, implemented in this paper.

In the present system, we speculate that it might be due to the relativistic character with strong spin-orbit coupling of the carriers in the system. Our observations suggest a detailed theoretical model of such systems is essential and it would have to contain ingredients of low dimensionality, relativistic electrons, localized magnetic moments and strong spin-orbit coupling. Further evolution of the Fermi surface with electrostatic gating and in-plane magnetic field can be explored which we plan to do in a follow-up study to build up a complete understanding of the mechanism.[57]”

[56] N.W. Ashcroft, and N.D. Mermin, Solid State Physics, Cengage Learning, ISBN 8131500527, 9788131500521, (2011).

[57] D. Bucheli, M. Grilli, F. Peronaci, G. Seibold, and S. Caprara, Phys. Rev. B 89, 195448 (2014).

4) Actually, except for the Fig. 4, the other three figures are simply the basic characterizations. I did not learn anything new from these results. Perhaps, these should be put into the supporting information. And the authors should focus only on the Figure 4 and expand the explanations, provide more solid evidence, fit their results with their model, etc.

We unequivocally disagree with the Referee. Data and results shown in figures 1-3 are crucial in establishing the quality and properties of the interface. For example RHEED shows the layer by layer epitaxial growth of the LVO films. The narrow rocking curve shows the high crystallinity of the sample and the Laue fringes obtained shows the high quality of the interface. The resistivity and Hall measurements ensure the 2D nature of the conduction. Infact the Referee’s point no. 2, regarding the interface quality raised in their earlier comment are all answered in the figures 1 to 3 of the manuscript. Additionally, the large SO coupling which is one of the main findings of this study, is also extracted from a fit in Fig. 3b.

However, we do take Referee’s concern seriously and to strengthen our analysis and new results, we have further analyzed our PHE and AMR data and have found that our interface also shows signature similar to that of ABJ chiral anomaly at low fields as seen in Dirac/Weyl semimetals. We have added complete analysis in the revised manuscript and also an additional figure (Fig. 5) showing the topological signatures in our LVO-KTO system. Thus, we are reporting a further novel finding in this interface.

Referee #2

The manuscript “Planar Hall effect and Anisotropic Magnetoresistance in a polar-polar interface of LaVO₃-KTaO₃ with strong spin-orbit coupling”, by Neha Wadehra, et al., seems to contain results that are sound and interesting enough to warrant publication.

We thank the Referee for appreciating our work and recommending publication. We respond to the Referee's specific points below.

Before a final decision is made, I ask the authors to consider the following comments and suggestions, to improve the quality of their presentation.

1. The list of evidences for emergent physical properties of Rashba systems, should be updated somewhat, after 2013. I found some Refs.

Sci. Rep. DOI: 10.1038/srep12751;

EPL, 112 (2015) 17004 www.epljournal.org doi: 10.1209/0295-5075/112/17004;

Nat. Mater. 15, 1261–1266 (2016);

Nat. Mater. 15, 1224–1225 (2016);

PRL 119, 256801 (2017);

but the authors may find other references too.

We thank the Referee for their comment. We have added the above and the below mentioned additional recent references in the revised manuscript.

- a) F. Zheng, L.Z. Tan, S. Liu, and M. Rappe, Nano Lett., 15, 7794 (2015).
- b) S.D. Stranks, and P. Plochocka, Nat. Mater., 17, 377 (2018).
- c) J. Puebla, F. Auvray, N. Yamaguchi, M. Xu, S.Z. Bisri, Y. Iwasa, F. Ishii, and Y. Otani, Phys. Rev. Lett. 122, 256501 (2019).

2. The concept of 2D and 3D resistivity is confusing, I rather suggest to discuss it in term of contributions parallel and perpendicular to the interface.

We thank the Referee for her/his comment. This confusing terminology has been removed and new statements added in an attempt to simplify the meaning of these resistivities. The following text is the modified text on page 2, column 2, and paragraph 2, line 5:

Figure 2(a) shows the temperature dependent resistivity for all the samples, where the upper panel shows the two dimensional resistivity (ρ_{2D}) for conduction parallel to the interface and the lower panel shows the three dimensional resistivity (ρ_{3D}) normalized by the film thickness. The 3 ml sample although conducting at room temperature exhibits an upturn near 30 K. All other samples with LVO more than 3 ml are conducting down to 1.8 K. In contrast to the wide range of values found for ρ_{3D} , the same data plotted as (ρ_{2D}) shows that the data for all conducting samples essentially collapse to a narrow range of (ρ_{2D}) values. This indicates that LVO film itself is indeed insulating and that only the interface forms the conducting channel.

3. As I understand, the authors did not attempt a study as a function of the carrier density, so the reader is left wondering whether multiband/multicarrier physics may play a role at these interfaces, like it does in LAO/STO (see, e.g., <https://doi.org/10.1038/s41563-019-0354-z>). I was wondering whether the authors can at least discuss this issue, or plan future studies. This issue is particularly relevant, because the authors measure the Hall effect up to high magnetic fields.

We thank the Referee for this valuable comment. The charge carrier density could be tuned using gating and as suggested by the Referee, it is definitely an interesting direction to go and explore. We are currently not set up to do this but we plan to do it in a separate study. We have also mentioned this in the revised manuscript.

Regarding Referee's concern about multiband conduction, we would like to mention that our Hall measurements show perfectly linear dependence of Hall resistance on applied magnetic field even upto 14 T magnetic field, ruling out any multiband or multicarrier effects where a non-linear behavior is expected. Considering Referee's point, we have now added the raw Hall data for the 4 ml sample in the inset of Fig. 2(c) and for all other samples in supplementary information. In addition, on the suggestion of the Referee we have now included a discussion on this in the revised manuscript.

We have added the following text on page 3, column 1, paragraph 1, line 13:

“The charge carrier density of the samples can in principle be tuned using electrostatic gating and multicarrier physics can be explored but we plan to do it in a separate study. In the present case the linear variation of Hall resistance with magnetic field confirms the single type of charge carrier in our system. Figure 2(b) inset shows the Hall resistance of 4 ml LVO-KTO. The Hall data for other samples is presented in supplementary information.”

4. Can the authors explain why there is no need for a Kohler contribution in Eq. (1)? Such a contribution is expected in high-mobility metallic systems (see, e.g., DOI: 10.1038/srep12751).

We thank the Referee for his comment. We would like to mention that Kohler term gives the B^2 orbital magnetoresistance contribution. Since we had fitted in the diffusive regime only, we had not used the Kohler term for fitting. But taking Referee's suggestion, we have also fitted our MR data using the full equation and would like to mention that the B_{50} value remains almost same. Fig. 3(b) in the revised manuscript now shows fitting using the full equation.

5. I found two occurrences of cm^2 rather than cm^{-2} for the units of the carrier density.

We thank the Referee for pointing out these typos which we have now corrected.

6. The effect of an external magnetic field on the Rashba band structure has been discussed in PRB 89, 195448 (2014).

We thank the Referee for pointing out this manuscript which we found to be relevant to our discussions. We have now included this in our references.

Following text is added in the manuscript on page 6, column 2, paragraph 1, line 1:

“In the present system, we speculate that it might be due to the relativistic character with strong spin-orbit coupling of the carriers in the system. Our observations suggest a detailed theoretical model of such systems is essential and it would have to contain ingredients of low dimensionality, relativistic electrons, localized magnetic moments and strong spin-orbit coupling. Further evolution of the Fermi surface with electrostatic gating and in-plane magnetic field can be explored which we plan to do in a follow-up study to build up a complete understanding of the mechanism.[57]”

Reference:

[57] D. Bucheli, M. Grilli, F. Peronaci, G. Seibold, and S. Caprara, Phys. Rev. B 89, 195448 (2014).

Once the authors have complied with the above issues, I suggest that their manuscript may be accepted for publication.

Referee #3

The authors report transport measurements in a new 2DEG stabilized at the KTO/LVO oxide interface, arising from a polar catastrophe analogous to the LAO/STO interface. The material quality looks very good given the RHEED oscillations indicating layer-by-layer growth, and the narrow rocking curve from XRD.

One of the major findings is the 4.4 T spin-orbit field determined from weak antilocalization fits. This shows that the spin-orbit coupling in KTO/LVO is the largest amongst oxide 2DEGs as seen from Fig. 3c. This is potentially of great interest to the spintronics community. It would have been interesting to know how this changes with gating.

We thank the Referee for appreciating our work and recognizing the high quality of the interface. The high quality of the interface has allowed us to make measurements which have revealed several novel behavior. The Referee has correctly pointed out that spin-orbit coupling of LVO-KTO system is highest among oxide 2DEGs and can further be enhanced using gating. However, we would like to mention that we are planning to check the effect of gating on the spin-orbit coupling in a separate study. This is beyond the scope of the present manuscript because of our restriction in the measurement system.

The authors also present interesting Planar Hall Effect (PHE) R_{xy} and Anisotropic Magnetoresistance (AMR) R_{xx} data as a function of the angle θ between the current flow direction and the in-plane field. At low fields (~ 3 T) they see the expected two-fold AMR going like $\cos^2(\theta)$, but at high fields (~ 14 T) it changes to a four-fold symmetry $\cos^4(\theta)$. Such

a term is allowed by the crystal symmetry, but it is not described by the theoretical modeling presented here. The authors do not comment on the angular dependence of the PHE, which retains two-fold symmetry at all fields.

Here are some ways in which the authors can improve their manuscript.

1) From the simple Rashba SOC model, the authors obtain the conventional $\cos^2(\theta)$ dependence in R_{xx} in the low magnetic field regime. However, the authors neglect to explain if this model correctly predicts the angular dependence for R_{xy} in this regime. The authors should discuss this.

We thank the Referee for raising this important point. We would like to mention that taking Referee's suggestion we have gone through our theoretical analysis again and have extended it to discuss the angular dependence of R_{yx} in this regime. Accordingly we have modified the text in the revised manuscript on page 5, column 2, paragraph 3, line 1:

“In our system, the degenerate energy parabola of electrons is Rashba split into two parabolas as shown in Fig. 4(e). Application of an external magnetic field in the conduction plane further adds a Zeeman splitting term. The external parabola is called the majority band and the internal parabola is called the minority band. Depending on the propagation vector k , spin of the electron, Rashba strength parameter α and the direction and magnitude of the external applied magnetic field, the electrons can make transitions between majority-to-majority (or minority-to-minority) i.e. intra-band transitions and majority-to-minority (or minority-to-majority) i.e. inter-band transitions. Each allowed transition results in back-scattering of the conduction electrons and hence, contributes to increase in resistance. The energy eigen values for the spin bands can be calculated by solving the Hamiltonian, which in the absence of the applied magnetic field, can be written [35]:

$$H = \epsilon(k) - \alpha(\sigma_x k_y - \sigma_y k_x) \quad (4)$$

Where, $\epsilon(k)$ is free electron energy, $\sigma(x; y)$ are the Pauli spin matrices, and k_x and k_y are the wave vectors in x and y direction. The electronic transition probability between the bands can be calculated using the eigen vectors for each band and finding the transition matrices. The eigen vectors used for the majority and minority bands are:

$$(1 + i e^{i\theta k})/\sqrt{2} \text{ and } (1 - i e^{i\theta k})/\sqrt{2} ,$$

respectively, where, θ is the angle between the k -vector and the x -axis. Figure 4(e) shows the Rashba energy-split bands (numbered as 1, 2, 3 and 4) with spin texture for a fixed energy value.

Now, imagine that a B-field is applied in a direction which is also coincident with the above x -axis (which is however distinct from the x -axis in the laboratory frame, i. e., the frame of the planar sample, along which the current is applied). The present xy -frame then defines the principal coordinate system, in the sense of Taskin et al. [33] The corresponding Zeeman term contains the x -component of the Pauli spin which does not affect the term proportional to k_y , in Eq. (4). However, the Zeeman coupling, being off-diagonal in the σ_y -representation, can cause a spin-flip thereby triggering a reversal in k_x . Interestingly, these momentum-reversal transitions are akin to

Drude scattering that is ever present, albeit small, as a ‘residual’ resistivity. [56] Now, when the B-field is not too large (< 8 T, in our experiments), the additional transition probabilities due to the Zeeman interaction can be calculated from the ‘Golden Rule’ of perturbation theory, thus yielding a quadratic dependence on B, which have to be supplanted to the residual Drude resistivity parallel to the x-axis. The perpendicular component however retains only the residual part, for reasons mentioned earlier.

The allowed transitions between different bands (T_{13} , T_{14} , T_{23} , T_{24}) having finite probability are shown with arrows in Fig. 4(e). These have been computed in the supplementary section and are presented in Fig. 4(f). With the parallel and perpendicular (‘diagonal’) components in hand we can transform back to the laboratory frame *a la* Taskin et al. [33] Since the residual Drude resistivity cancels out from the difference between the parallel and perpendicular components, the dominant contribution to the resistivity arises from the Rashba effect. Further, following Taskin et al. [33], $R_{XX} \sim (\cos^2 \varphi)$, while $R_{YX} \sim (\cos \varphi \cdot \sin \varphi)$, both being proportional to $(B)^2$, in conformity with our data shown in Fig. 5 below. Here, φ is the angle between the applied magnetic field and the direction along which the current is measured. As we mentioned earlier, beyond 8 T, there is a substantial departure in the $(\cos^2 \varphi)$ -behavior of R_{XX} , as well as its $(B)^2$ dependence, a theoretical understanding of which requires going beyond the Golden Rule of perturbation theory and perhaps also beyond the simple Rashba effect, implemented in this paper.

In the present system, we speculate that it might be due to the relativistic character with strong spin-orbit coupling of the carriers in the system. Our observations suggest a detailed theoretical model of such systems is essential and it would have to contain ingredients of low dimensionality, relativistic electrons, localized magnetic moments and strong spin-orbit coupling. Further evolution of the Fermi surface with electrostatic gating and in-plane magnetic field can be explored which we plan to do in a follow-up study to build up a complete understanding of the mechanism.[57]”

[56] N.W. Ashcroft, and N.D. Mermin, Solid State Physics, Cengage Learning, ISBN 8131500527, 9788131500521, (2011).

[57] D. Bucheli, M. Grilli, F. Peronaci, G. Seibold, and S. Caprara, Phys. Rev. B 89, 195448 (2014).

2) Looking at the calculation for $R_{xx}^{\{norm\}}$, I feel that there may be a minor mistake. From Fig. 4a, it seems that $R_{xx}^{\{symm\}} \sim 372$ Ohms. The value of R_0 (R_{xx} at 0 degrees) is ~ 376 Ohms. So, $R_{xx}^{\{norm\}}$ at 0 degrees would be $(376-372)/376 = 0.01$. However, in Fig. 4c $R_{xx}^{\{norm\}}$ changes from -1 to 1 as labeled in the color scale? Perhaps the corrected equation is $R_{xx}^{\{norm\}} = (R - R_{xx}^{\{symm\}})/(R_0 - R_{xx_}\{symm\})$

We agree with the Referee and thank her/him for pointing out this important point. We have corrected the equation in the revised manuscript.

3) Since R_{xx} is dimensionless, it should not have units of Ohms in the color bar Fig. 4c.

We thank the Referee for her/his comment. We have made correction in the figure in the revised manuscript.

I feel that the modeling should be strengthened, but the experimental data at the core of the paper is compelling even if the field variation of the AMR symmetries is not yet well understood. On balance, I am happy to recommend publication in Nature Communications once the above questions are addressed.

We thank the Referee for the positive recommendation. We understand that the modeling is qualitative at the moment and we thank the Referee for stating that our experimental results on their own are compelling enough. The modeling is our attempt at reaching at least a qualitative and semi-quantitative understanding of the observed novel behavior. In the revised manuscript we have included further analysis of the R_{xx} and R_{yx} and shown that it might have a substantial chiral contribution. This is an additional novel observation in our materials.

We hope that we have responded satisfactorily to the comments of the Referees and have been able to re-emphasize the novel results discovered in this work. These include the largest SO coupling in an oxide 2DEG, anomalous magnetotransport behavior like AMR and PHE which are not expected in normal oxide interfaces, chiral contributions in the magnetoresistance suggesting novel topological properties arising from the strong SO and Rashba effects. We are sure that our novel results will spawn further experimental and theoretical studies to understand the novel behavior observed in this work.

Reviewers' comments:

Reviewer #1 (Remarks to the Author):

In the revised manuscript, the authors had addressed a few concerns, however, I don't think they really satisfied my questions clearly. Here is my answer,

1) Indeed there are a few groups had published manuscripts on the magnetic 2DEG systems. The SO actually similar to the current report. The mechanism is similar to this manuscript so that makes this work loses the novelty. Especially the use of Ta-V, this kind of 3d-5d interfaces. The results are based on a speculated explanation and did not provide more solid evidence to support their speculations.

2) I agree figures 2 and 3 may keep as it is, but can be combined together. These are the raw transport measurements and analysis of the sample. Figure 1, even though important, doesn't provide much information, should be concluded in the SI. The successful growth of LVO films had been reported earlier.

3) I insist the authors should provide two important characterizations: firstly, XAS to prove the valence states of V as the V has multiple states. A tiny change on the doping electrons would modify the spin states of V. The exact and correct electronic state of V is fairly important for the explanation. Secondly, the STEM characterizations. For the XRD, I can only conclude the LVO films might have good crystallinity. However, since the properties they claimed are based on the interfaces. The chemical intermixing, sharpness and continuity are important and basis for the discussion.

4) The theoretical model are completely different from the earlier version. Again, it is a speculation. Based on the quick and fewer transport data at a fixed temperature, I doubt the validity of the model. The authors should provide more consistent characterizations and proofs of their model.

In conclusion, I think the authors need more basic characterizations and detailed analysis of their data. Before that, I can not recommend it to be published in Nat. Commun.

Reviewer #2 (Remarks to the Author):

I find that the revised version of the manuscript "Planar Hall Effect and Anisotropic Magnetoresistance in a polar-polar interface of LaVO₃-KTaO₃ with strong spin-orbit coupling", by Neha Wadehra, et al., has been significantly improved by the authors, following the suggestions and remarks of all the referees.

It is evident that some aspects remain controversial, but in my opinion the authors' reply to the various criticisms, while not always compelling, is sufficiently sound and not misleading. Moreover, I think that the subject is still relevant and worth being discussed within the scientific community, the results are reasonable and are presented in a fair manner, highlighting the controversial points and the need for further studies. Thus, it can be expected that this piece of work may trigger future work in this research field.

For all the above reasons, I suggest that this manuscript may now be accepted for publication.

Reviewer #3 (Remarks to the Author):

I have gone through the revised manuscript and to the authors' response to my report and to the reports of the other two referees. I am satisfied with the changes the authors have made and their replies. I recommend the paper for publication.

We have gone through the reviewer report very carefully. We are happy to see that Reviewers 2 and 3 have accepted our work in the present form for publication in Nature Communications.

Reviewer 1 has raised two concerns.

We wish to point out that these two concerns raised by Reviewer 1 are fresh issues of no direct relevance to the revisions carried out by us in the last stage. This is somewhat unusual, in the sense that the request for such time-consuming measurements could very well have been made at an earlier stage. Nevertheless, we have considered the recommendations as they stand, and tried to respond to them to the best of our ability.

1. The first concern is about the oxidation state of Vanadium in the LaVO_3 film. We find this might be a technically relevant issue, since it is likely to further elucidate our understanding of the physical mechanism leading to the interface conductivity. We have therefore performed photo-emission spectroscopy of LaVO_3 film and included this data in this final revised manuscript.

2. The second concern is about the structural quality of the interface. We had already provided the following measurements that strongly indicate (albeit indirectly) an atomically well-defined interface,

- A. RHEED intensity oscillations observed throughout the growth process strongly suggest a layer by layer growth.

- B. Laue fringes present in the x-ray diffraction data also suggest an almost atomically sharp interface.

- C. We have shown that the interface conductivity appears only when the LaVO_3 film thickness exceeds 3 monolayers. This clearly indicates that the interface conductivity originates from the polar catastrophe mechanism and cannot be ascribed to atomic intermixing at the interface.

The STEM measurement suggested by the Reviewer could be useful to find out whether the interface is atomically sharp at the monolayer level or not. However, such information will be only of academic value and will not make any meaningful contribution to our main result or its understanding, particularly in view of the enormous time and effort involved in obtaining time at an atomic resolution TEM facility (not available on site) and preparation of cross-sectional samples.

We sincerely hope that with this modification of our manuscript will be accepted for the publication.